# SARS-CoV-2 nsp14 Exoribonuclease Removes the Natural Antiviral 3′-Deoxy-3′,4′-didehydro-cytidine Nucleotide from RNA

**DOI:** 10.3390/v14081790

**Published:** 2022-08-16

**Authors:** Nicholas H. Moeller, Kellan T. Passow, Daniel A. Harki, Hideki Aihara

**Affiliations:** 1Department of Biochemistry, Molecular Biology, and Biophysics, University of Minnesota, Minneapolis, MN 55455, USA; 2Institute for Molecular Virology, University of Minnesota, Minneapolis, MN 55455, USA; 3Masonic Cancer Center, University of Minnesota, Minneapolis, MN 55455, USA; 4Department of Medicinal Chemistry, University of Minnesota, Minneapolis, MN 55455, USA

**Keywords:** SARS-CoV-2, nsp14, proofreading, chain terminator, ddhCTP, RNA-dependent RNA polymerase, nucleoside analogue, antiviral drug, exoribonuclease, nucleotide

## Abstract

The on-going global pandemic of COVID-19 is caused by SARS-CoV-2, which features a proofreading mechanism to facilitate the replication of its large RNA genome. The 3′-to-5′ exoribonuclease (ExoN) activity of SARS-CoV-2 non-structural protein 14 (nsp14) removes nucleotides misincorporated during RNA synthesis by the low-fidelity viral RNA-dependent RNA polymerase (RdRp) and thereby compromises the efficacy of antiviral nucleoside/nucleotide analogues. Here we show biochemically that SARS-CoV-2 nsp14 can excise the natural antiviral chain-terminating nucleotide, 3′-deoxy-3′,4′-didehydro-cytidine 5′-monophosphate (ddhCMP), incorporated by RdRp at the 3′ end of an RNA strand. Nsp14 ExoN processes an RNA strand terminated with ddhCMP more efficiently than that with a non-physiological chain terminator 3′-deoxy-cytidine monophosphate (3′-dCMP), whereas RdRp is more susceptible to chain termination by 3′-dCTP than ddhCTP. These results suggest that nsp14 ExoN could play a role in protecting SARS-CoV-2 from ddhCTP, which is produced as part of the innate immune response against viral infections, and that the SARS-CoV-2 enzymes may have adapted to minimize the antiviral effect of ddhCTP.

## 1. Introduction

SARS-CoV-2 replication depends on the exoribonuclease (ExoN) activity of non-structural protein 14 (nsp14), which provides proofreading during viral RNA synthesis and antagonism against host innate immune responses [1,2,3,4,5]. In the proofreading function, nsp14 in complex nsp10, which stabilizes the nuclease-active conformation of the N-terminal ExoN domain of nsp14 [6,7,8,9], removes nucleotides misincorporated by the low-fidelity RNA-dependent RNA polymerase (RdRp) to facilitate the faithful replication of the large (~30 kb) coronavirus genome [10,11,12,13,14]. In addition to correcting replication errors, the ExoN activity of the nsp14–nsp10 complex also excises nucleotide analogues misincorporated into RNA by RdRp and thereby compromises their antiviral efficacy. For instance, remdesivir, a broad-spectrum antiviral drug effective against SARS-CoV-2, is readily incorporated during RNA synthesis by SARS-CoV-2 RdRp (nsp12) but can be removed from the RNA strand by nsp14-ExoN [6]. Consistently, a mutant of murine hepatitis virus (MHV), a betacoronavirus closely related to SARS-CoV-2 with inactivated ExoN proofreading, was shown to be significantly more sensitive to remdesivir [15].

Nucleoside analogues are widely used as antiviral drugs. Besides remdesivir mentioned above, this class of drugs include AZT targeting HIV-1 reverse transcriptase, sofosbuvir targeting hepatitis C virus RNA polymerase, and ganciclovir/aciclovir targeting herpes virus DNA polymerases [16,17,18]. These compounds are misincorporated in the nucleotide triphosphate form by viral polymerases and act as chain-terminators to abort RNA/DNA synthesis. A distinct class of nucleoside analogues has also been developed, which acts through lethal mutagenesis rather than the chain-termination mechanism and shows a high genetic barrier to resistance [19,20,21,22,23]. Molnupiravir, a broad-spectrum antiviral that is an orally bioavailable prodrug of the nucleoside analogue β-D-N4-hydroxycytidine (NHC), is misincorporated by SARS-CoV-2 RdRp as its 5′-triphosphate metabolite and elicits mutations due to promiscuous base-pairing [24,25]. Although work using MHV suggested that NHC evades or overcomes the viral proofreading activity, biochemical studies showed that SARS-CoV nsp14-ExoN can excise one of such mutagenic nucleoside analogues, ribavirin, from the 3′ end of an RNA strand [19,26].

While the chain-terminating and mutagenic nucleoside analogues mentioned above are human-made drugs, recent studies have identified a novel antiviral chain-terminating nucleotide, 3′-deoxy-3′,4′-didehydro-cytidine triphosphate (ddhCTP, Figure 1), produced in mammalian cells by the activity of an interferon-inducible enzyme Viperin/RSAD2 [27,28]. Viperin is a radical S-adenosyl methionine (SAM)-dependent protein and has also been found in prokaryotes, where it produces ddhCTP, ddh-guanosine triphosphate (ddhGTP), and ddh-uridine triphosphate (ddhUTP) to provide protection against phage infections [29,30]. ddhCTP was shown to be misincorporated by RdRp from multiple members of the flaviviral genus and it inhibits replication of Zika virus in vivo [27,31]. The sera of patients with viral infections, including COVID-19, were found to have elevated levels of the nucleoside form of ddhCTP (ddhC), highlighting a diagnostic potential for ddhC [32]. Furthermore, a prodrug form of ddhCTP has been developed as a chemical strategy to adapt this natural antiviral mechanism [33]. However, even though ddhCTP is misincorporated during RNA synthesis by SARS-CoV-2 RdRp in vitro, it does not affect SARS-CoV-2 replication in cells [4]. A possible explanation for these observations is that nsp14-ExoN proofreading activity protects SARS-CoV-2 replication by excising incorporated ddhCMP from the nascent RNA. In this study, we examined biochemical activity of SARS-CoV-2 nsp14-ExoN to excise ddhCMP from the 3′ end of an RNA strand to help address this question.

## 2. Materials and Methods

ddhCTP was synthesized as reported previously [33]. Data demonstrating its purity are shown in Appendix A. Ribonucleotide triphosphates (NTPs: ATP, CTP, GTP, UTP) and 3′-deoxy-cytidine 5′-triphosphate (3′-dCTP) were purchased from New England Biolabs and TriLink Biotechnologies, respectively. SARS-CoV-2 nsp7, nsp8, nsp10, nsp12, and nsp14 were recombinantly expressed in *Escherichia coli* and purified as reported previously [6]. RNA oligonucleotides were synthesized by Integrated DNA Technologies. TBE-Urea polyacrylamide gels (Invitrogen, Novex) were purchased from ThermoFisher Scientific.

RNA primer extension reactions shown in Figure 2 were performed with 750 nM 5′-fluoroscein (FAM)-labeled 20-nucleotide (nt) primer annealed to a 3′-protected (2′,3′-dideoxy-terminated) 30-nt template, 2.0 μM nsp7/8 complex, 1.0 μM nsp12, and 7.5 μM each of indicated nucleotides, in 20 mM HEPES-NaOH, pH 7.5, 5.0 mM MgCl_2_, 10 mM dithiothreitol, and 0.01% Tween-20. After an initial incubation at 37 °C for 25 min, the reactions were supplemented with 200 μM NTPs or water (as controls) and further incubated for 25 min to let RdRp re-extend stalled products. The reactions were stopped by the addition of formamide to 89% and heating to 95 °C for 10 min and the products were separated by gel electrophoresis on a 15% polyacrylamide TBE-Urea denaturing gel, which was scanned on a Typhoon FLA 9500 imager. For the primer extension reactions shown in Figure 3, the same primer as above annealed to a 3′-protected 40-nt template was extended in the presence of 15 μM CTP, 60 μM ATP, 45 μM GTP, 30 μM UTP, and varying concentrations (0 to 1.5 mM) of 3′-dCTP or ddhCTP at 37 °C for 25 min in the same reaction condition as above. The reaction products were separated by gel electrophoresis and visualized by scanning for fluorescence as described above. For both the 30-nt (Figure 2) and 40-nt (Figure 3) templates, the full extension product ran as two discrete bands, which was due to incomplete strand separation despite the denaturing condition used and heating prior to gel-loading (the residual double-stranded form is denoted by # in Figure 2 and Figure 3). For the analysis in Figure 3c, band intensities were quantitated using ImageJ (https://imagej.nih.gov/ij/, accessed on 14 August 2022). For each condition, the sum of fluorescence intensities from the single and double-stranded full-length product bands was divided by the total fluorescence signal integrated over the lane to calculate the fraction of full-length product.

Exonucleolytic degradation reactions shown in Figure 4 used the stalled/abortive products of RdRp reactions (Figure 2, lanes 7–9) as the starting RNA substrate. Following the initial primer extension in the presence of GTP and CTP/3′-dCTP/ddhCTP, RpRp was inactivated by heating to 95 °C for 10 min. The samples were cooled on ice, supplemented with 10, 40, or 160 nM (final concentrations) of nsp14–nsp10 complex, and incubated at 37 °C for 10 min. The reaction products were separated by gel electrophoresis and bands visualized as described above.

## 3. Results

### 3.1. SARS-CoV-2 RdRp Incorporates ddhCTP

We prepared SARS-CoV-2 RdRp by mixing purified nsp12, the polymerase catalytic subunit, with its key co-factors nsp7 and nsp8, which confer processivity to nsp12 [34,35]. The reconstituted RdRp complex extended a 5′-FAM labeled 20-nt primer annealed to an unlabeled, 2′,3′-dideoxy-terminated 30-nt template into the 30-nt product in the presence of NTPs (Figure 2, lanes 1, 2), consistent with the primer-dependent RNA polymerase activity of coronavirus nsp12 reported previously [36]. As expected from the nucleotide sequence of the template (Figure 2a), CTP alone did not support extension of the primer whereas GTP alone allowed extension by single nucleotide (Figure 2, lanes 3, 4). A reaction with GTP and CTP allowed primer extension by two nucleotides (Figure 2, lane 7), and the product stalled after CMP incorporation was readily converted to the 30-nt product by re-extension with RdRp in the presence of the full set of NTPs (Figure 2, lane 10).

Earlier studies have shown that SARS-CoV-2 RdRp can incorporate the natural chain-terminating nucleotide ddhCTP during RNA synthesis [4]. To confirm that SARS-CoV-2 RdRp can utilize ddhCTP as a substrate, we performed the primer extension reaction with ddhCTP, or 3′-dCTP as a control, substituted for CTP. Whereas either ddhCTP or 3′-dCTP alone did not support primer extension (Figure 2, lanes 5, 6), the reaction with GTP and either ddhCTP or 3′-dCTP generated a product that migrated more slowly in gel than the stalled product generated with GTP alone, which suggested extension by two nucleotides (Figure 2, lanes 8, 9). Although these products migrated slightly faster than that obtained with GTP and CTP (Figure 2, lane 7), they could not be re-extended in the presence of natural NTPs, confirming the presence of a chain-terminating nucleotide at the 3′ terminus (Figure 2, lanes 11, 12). The results collectively demonstrate that ddhCTP and 3′-dCTP can be incorporated by SARS-CoV-2 RdRp to abort RNA synthesis.

**Figure 2 viruses-14-01790-f002:**
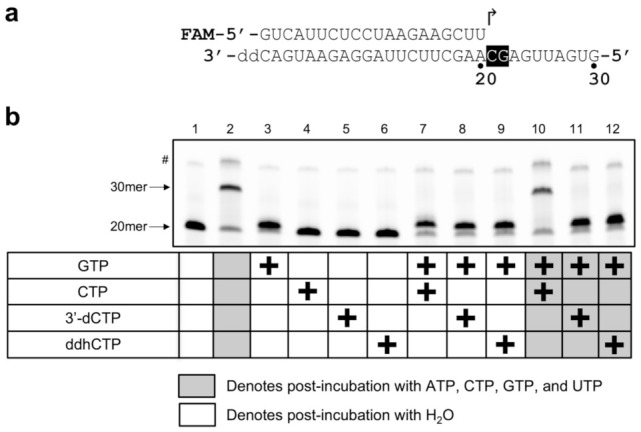
Incorporation of 3′-dCTP or ddhCTP by SARS-CoV-2 RdRp. (**a**) Sequences of the 5′-fluorescein-labeled 20-nt RNA primer and the 3′-protected 30-nt RNA template. (**b**) Products of primer extension by RdRp in the presence of the nucleotides indicated by “+” during the first incubation, followed by re-extension in the presence (shaded columns) or absence (non-shaded columns) of all NTPs. Whereas the product stalled after CTP incorporation (lane 7) is readily re-extended into the 30-nt product (lane 10), those stalled after 3′-dCTP or ddhCTP incorporation are chain-terminated and therefore could not be re-extended (lanes 11 and 12). Residual double-stranded form of RNA is denoted by #. A representative gel result for 3 independent experiments is shown.

### 3.2. ddhCTP Is a Less Preferred Substrate than 3′-dCTP for SARS-CoV-2 RdRp

Next, we tested primer extension in the presence of ddhCTP or 3′-dCTP in addition to the normal NTPs to examine how well these chain-terminating nucleotides compete with CTP for incorporation by RdRp. For this experiment, we used the same 5′-FAM labeled primer as above, annealed to a 40-nt template strand. In a titration of 3′-dCTP concentration from 0 to 1.5 mM against a fixed concentration (15 μM) of CTP (Figure 3), we observed progressive reduction of the fully extended (40-nt) product and corresponding accumulation of abortive extension products. A similar titration of ddhCTP also generated abortive products at higher chain terminator concentrations, although the extent of incorporation was less compared to 3′-dCTP. Internal chain termination was apparent at the lowest 3′-dCTP/CTP molar ratio of 12.5 but not for the same ddhCTP/CTP ratio (Figure 3b, lanes 5 vs. 7), and generation of the fully extended product was reduced by 32.8 and 11.2% in the presence of the highest concentration (1.5 mM) of 3′-dCTP and ddhCTP, respectively (Figure 3b, lanes 2 vs. 10; Figure 3c; Appendix A). These results suggest that ddhCTP is a less preferred substrate than 3′-dCTP for SARS-CoV-2 RdRp, consistent with earlier single-molecule studies that showed more frequent incorporation of 3′-dCTP than ddhCTP by SARS-CoV-2 RdRp [4].

**Figure 3 viruses-14-01790-f003:**
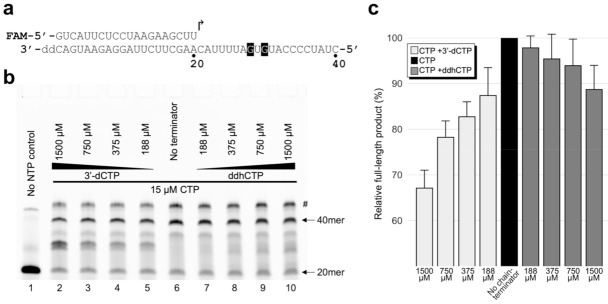
Incorporation of 3′-dCTP or ddhCTP by SARS-CoV-2 RdRp in the presence of CTP. (**a**) Sequences of the 5′-fluorescein-labeled 20-nt RNA primer and the 3′-protected 40-nt RNA template. The arrow indicates the direction of primer extension. (**b**) Products of primer extension by RdRp in the presence of the indicated concentrations of 3′-dCTP (lanes 2–5) or ddhCTP (lanes 7–10) in addition to 15 μM CTP. Lane 6 shows extension with no chain terminator. Residual double-stranded form of the fully extended 40-nt product is denoted by #. A representative gel result for 3 independent experiments is shown. (**c**) Quantitation of the gel result in (**b**) and its replicates. Fraction of the fully extended product relative to that for the no-chain terminator control is plotted (see Appendix A for representative spectrograms). The means +/− standard deviation (*n* = 3) are shown.

### 3.3. SARS-CoV-2 nsp14 ExoN Excises ddhCMP More Efficiently than 3′-dCMP

The ExoN activity of SARS-CoV-2 nsp14 has been shown to be able to excise various nucleotide analogues including ribavirin, remdesivir, and sofosbuvir from the 3′ terminus of RNA [6,26]. To test whether it can also excise ddhCMP incorporated into RNA, we incubated the stalled or chain-terminated 22-nt RNA generated in the primer extension reactions described above (Figure 2, lanes 7–9) with the purified nsp14–nsp10 complex at several different concentrations. The RNA strand with CMP at the 3′ terminus was most readily degraded, as expected (Figure 4, lanes 1–4). The RNA strand terminated with ddhCMP was processed almost as efficiently, although it showed less degradation with the lowest (10 nM) enzyme concentration (Figure 4, lanes 9–12). Notably, the RNA strand terminated with 3′-dCMP was more resistant than that with ddhCMP, consistent with the reported biochemical observations and structural explanation that removal of the 3′-OH group from RNA confers significant resistance against degradation by ExoN (Figure 4, lanes 5–8) [13]. Thus, SARS-CoV-2 nsp14 ExoN excises ddhCMP more efficiently than it does 3′-dCMP, which is a reversed trend in comparison to the preference of RdRp toward their triphosphate forms.

**Figure 4 viruses-14-01790-f004:**
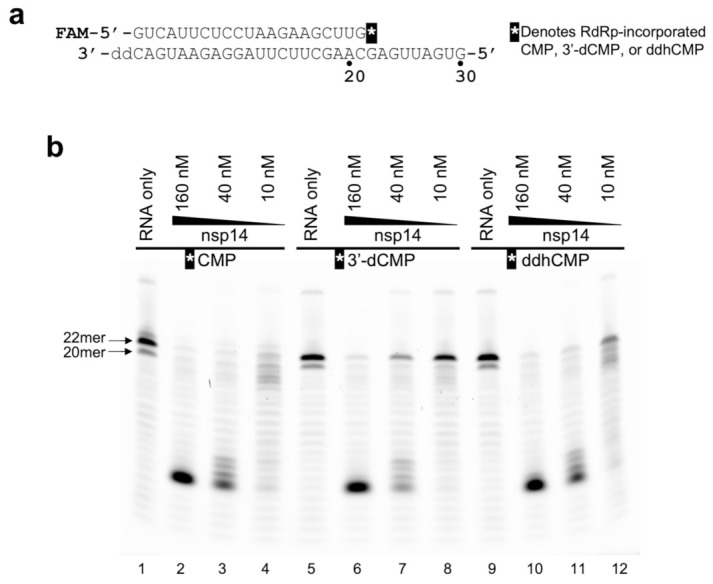
Degradation of 3′-dCMP or ddhCMP-terminated RNA strand by SARS-CoV-2 nsp14–nsp10 complex. (**a**) Sequences of the 22-nt RNA strand with CMP, 3′-dCMP, or ddhCMP at the 3′-terminus (stalled after the incorporation of CTP, 3′-dCTP, or ddhCTP by RdRp, respectively) and the 3′-protected 30-nt RNA template. (**b**) Degradation of the 22-nt RNA with CMP (lanes 1–4), 3′-dCMP (lanes 5–8), or ddhCMP (lanes 9–12) at the 3′-terminus by the SARS-CoV-2 nsp14–nsp10 complex at indicated concentrations. A representative gel result for 3 independent experiments is shown.

## 4. Discussion

The recent discovery of a natural chain-terminating nucleotide ddhCTP, produced upon viral infections, poses an important question as to how RNA viruses may overcome this endogenous antiviral mechanism in mammalian cells [27]. Our experiments show that the unique proofreading activity of the SARS-CoV-2 nsp14–nsp10 complex can excise ddhCMP incorporated at the RNA 3′ terminus, suggesting its role in viral antagonism. We found that nsp14-ExoN removed ddhCMP more efficiently than 3′-dCMP, which is particularly interesting given that 3′-dCMP has a smaller structural perturbation than ddhCMP from the normal cytidine nucleotide. On the other hand, our experiments as well as earlier reports showed that ddhCTP is less favorably incorporated than 3′-dCTP by SARS-CoV-2 RdRp. These observations combine to suggest that the SARS-CoV-2 replicase enzymes may have adapted to minimize the antiviral effect of ddhCTP by avoiding its incorporation and facilitating removal, if incorporated.

Coronavirus nsp14 is a multi-functional enzyme, which contains the N-terminal ExoN domain responsible for proofreading during RNA synthesis and the C-terminal guanine N7 methyltransferase domain essential in viral mRNA capping [7]. These enzymatic activities were also shown to be critical in the suppression of type I interferon (IFN-I) antiviral responses through translational shutdown, lysosomal degradation of the IFN-I receptor, and degradation of dsRNAs that serve as pathogen-associated molecular patterns [1,5,37,38,39]. Consistently, although RSAD2 that encodes for Viperin responsible for catalyzing the conversion of CTP to ddhCTP is an interferon-stimulated gene (ISG) and is highly upregulated in SARS-CoV-2 infected cells [40], its mRNA translation is significantly delayed along with other inefficiently translated ISGs [41]. Thus, it is possible that the production of ddhCTP itself is controlled by the activity of nsp14. SARS-CoV-2 nsp14 ExoN activity may thereby counter the antiviral effect of the natural chain-terminating nucleotide at two distinct steps and contribute to the evasion of hosts’ innate immune responses.

## Figures and Tables

**Figure 1 viruses-14-01790-f001:**
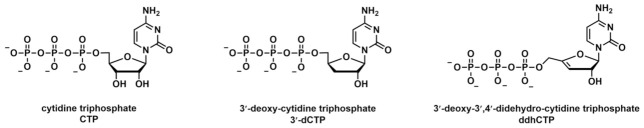
Chemical structures of CTP, and its chain-terminating analogues, 3′-dCTP and ddhCTP.

## Data Availability

Not applicable.

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
