# Peer review of "SARS-CoV-2 nsp14 Exoribonuclease Removes the Natural Antiviral 3′-Deoxy-3′,4′-didehydro-cytidine Nucleotide from RNA"

_viruses, 2022, doi:10.3390/v14081790_

Round 1

Reviewer 1 Report

Moeller et al, “SARS-C0V-2 nsp14 exoribonuclease removes the natural antiviral 3’-deoxy-3’,4’-didehydro-cytidine nucleotide from RNA”

The ExoN activity of the SARS-CoV-2 nsp14 protein removes nucleotides mis-incorporated during RNA synthesis.  The authors report that SARS-CoV-2 nsp14 can excise the naturally produces antiviral, chain-terminating nucleotide, ddhCTP, as well as ddCTP, incorporated at the 3’ terminus of RNA by RdRp.  These results suggest that nsp14 ExoN activity could play a role in protecting the virus from ddhCTP produced by the immune system. 

Overall, the manuscript is very well written and the figures are clear and concise.  The conclusions based on the data appear to be sound.  This work will be of interest to those focused on innate immunity, SARS-CoV-2, and antiviral mechanisms. 

The only significant comment is that for the sake of rigor and reproducibility more information regarding the analysis of the synthesized ddhCTP should be provided in the supplemental information.  At a minimum, it should include the purity of the product and any analytical chemistry results (NMR, mass spectrometry, etc) that confirm its structure.   This is important given the comparison of incorporation and excision between ddCTP and ddhCTP.

Reviewer 2 Report

Brief Summary

In this paper, Moeller et al. embark on a journey to explore sars-cov-2 virus ability to combat natural antivirals. Specifically, first they focus on the nsp12 ability to incorporate 3'-deoxy-3',4'-didehydro-cytidine nucleotide. Then, they follow the ability of nsp14 to excise the same nucleotide. 

Significance

COVID19 pandemic has devastated the world, with many millions of deaths. New variants of SARS-COV-2 continue to infect millions around the world, creating an urgent need for new antivirals. While the viral protein nsp14 exoribonuclease is known for mediates proofreading for the viral RNA replication, it has been shown to also excise replication inhibiting nucleotide analogue drugs. As the nsp14 viral activity poses a threat to both existing and future antivirals, we must understand every detail of its function spanning from molecular structure, biophysics, and biochemistry. Furthermore, knowledge gained about nsp14 could serve to develop new drugs to target the different parts of the viral replication and even nsp14 itself. 

Recommendations: 

I recommend accepting this paper with minor revisions for publication at the Viruses Journal. I am listing below minor suggestions for clarifying the science described in this paper. In case there are repeats of the biochemical assays I am not recommending any additional experiments, and I ask the authors to include the repeats in quantifications. If assays were not performed in duplicates or triplicates, I strongly recommend repeating the experiments.

General remark:

-       I noticed your gel figure legend do not mention “representative gel”. In figure 3c, where are the error bars? I assume nucleotide incorporation, extension, and excision assays were repeated. If so, please add error bars to quantification of gels. If not, please repeat assays, and add the repeats to the quantification. 

-       Please add to discussion or introduction the primer dependance of nsp12, and references, such as: 

o   “The RNA polymerase activity of SARS-coronavirus nsp12 is primer dependent “ Te Velthuis et al, 2010, NAR.  

o   “Biochemical characterization of a recombinant SARS coronavirus nsp12 RNA-dependent RNA polymerase capable of copying viral RNA templates”, Ahn et al., Arch Virol. 2012; 157(11): 2095–2104.

-       Please elaborate on nso12 and nsp14 structures in discussion or introduction, and reference, such as: 

o   Crystal structure of SARS-CoV-2 nsp10 bound to nsp14-ExoN domain reveals an exoribonuclease with both structural and functional integrity”, Lin, Nucleic Acids Research, 2021, Vol. 49, No. 9. 

-       If you are repeating assays, I strongly recommend adding RNA/DNA ladders to your gels, which I did not see in the original gel images attached. 

Comments:

Abstract: I recommend starting with a sentence about sars-cov-2 pandemic, devastation, or therapeutics. 

Line 86: “3'-protected”, please describe which 3’ protection. 

Line 116: Please add a brief description of the viral proteins nsp7, nsp8, nsp10, and nsp12. A good place would be in the beginning of the results, when you mention their addition to the reaction mixture.

Line 173: Please add a brief description of the viral proteins nsp10, after mention of their addition to the reaction mixture.

Line 182: “.. comparison to the preference of RdRp toward their trinucleotide forms.” Please clarify, such as “comparison to the preference of RdRp toward the three Cytidine forms,” or “.. CMP variants”.

Figure 2b: What is the top band? Is that the 20mer primer + 30mer template? If yes, please add this label to the figure, directly by the band as you did for the other bands, or designate with a star and a clarification in the legend as done for figure 3. 
